# Functional Assessments of Gynecologic Cancer Models Highlight Differences Between Single-Node Inhibitors of the PI3K/AKT/mTOR Pathway and a Pan-PI3K/mTOR Inhibitor, Gedatolisib

**DOI:** 10.3390/cancers16203520

**Published:** 2024-10-17

**Authors:** Aaron Broege, Stefano Rossetti, Adrish Sen, Arul S. Menon, Ian MacNeil, Jhomary Molden, Lance Laing

**Affiliations:** 1Celcuity, Inc., 16305 36th Ave N, Suite 100, Minneapolis, MN 55446, USA; abroege@celcuity.com (A.B.); asen@celcuity.com (A.S.); imacneil@celcuity.com (I.M.); jmolden@celcuity.com (J.M.); 2Department of Molecular and Cell Biology, University of California, Berkeley, CA 94720, USA; amenon@celcuity.com; 3College of Computing, Data Science, and Society, University of California, Berkeley, CA 94720, USA

**Keywords:** PI3K/AKT/mTOR pathway, gedatolisib, ovarian cancer, endometrial cancer

## Abstract

The frequent activation of the PI3K/AKT/mTOR (PAM) pathway makes it an attractive therapeutic target in gynecological cancers. Many PAM inhibitors selectively target single PAM pathway nodes, which can lead to reduced efficacy and increased drug resistance. In addition, compensatory pathways can be activated when only the PAM pathway is inhibited. Here, we show that gedatolisib, a PAM inhibitor targeting multiple PAM pathway nodes, exerted greater growth-inhibitory effects relative to single-node PAM inhibitors in gynecologic cancer cell models. In addition, gedatolisib combined with inhibitors of compensatory pathways involved in the estrogen response and cell cycle progression inhibited tumor growth in endometrial and ovarian cancer mouse models. Gedatolisib in combination with other therapies has previously shown promising preliminary clinical efficacy and safety in various solid tumor types. The non-clinical data presented here support the development of gedatolisib in combination with hormonal therapy and/or cell cycle inhibitors for gynecologic cancer treatment.

## 1. Introduction

Gynecologic cancers, such as endometrial cancer (EC), ovarian cancer (OC), and cervical cancer (CC), are frequently characterized by the dysregulation of the PI3K, AKT, and mTOR (PAM) pathway. Due to its key role in tumorigenesis, tumor progression, and drug resistance, the PAM pathway represents a promising target for gynecologic cancer therapy [1,2,3,4].

The PAM pathway controls several cellular functions, including metabolic homeostasis, biomolecule synthesis, cell survival, and cell proliferation [5,6,7]. Key nodes of this pathway include class I PI3K enzymes, AKT, and mTOR, which can be part of two distinct molecular complexes, mTORC1 and mTORC2 (Figure 1A). Class I PI3K enzymes are kinases consisting of a catalytic subunit (with four isoforms, p110α, β, γ, and δ, encoded by *PIK3CA*, *PIK3CB*, *PIK3CG*, and *PIK3CD*) and a regulatory subunit (also with different isoforms, including p85α and p85β, encoded by *PIK3R1* and *PIK3R2*). PI3K is activated in response to specific extracellular signals (e.g., growth factors, hormones, and nutrients) through multiple cell surface receptors, including G-protein-coupled receptors (GPCRs) and receptor tyrosine kinases (RTKs). Once activated, PI3K phosphorylates phosphatidylinositol (4,5)-bisphosphate (PIP2) and converts it into phosphatidylinositol (3,4,5)-trisphosphate (PIP3). In turn, PIP3 accumulation initiates a signaling cascade involving multiple effectors. One of the main effectors of PI3K products, activated PDK1 and PIP3, is AKT (with three different isoforms, AKT1, AKT2, and AKT3), which controls the function of many downstream targets, including mTORC1. AKT is also phosphorylated and activated by mTORC2, another downstream PI3K effector that provides an additional layer of control over increased AKT activity. The conversion of PIP3 to PIP2 by the PTEN phosphatase represents one of the main repressing, termination mechanisms of the PAM signaling pathway [5,6,7].

Dysregulation of the PAM pathway in cancer can be due to multiple factors. An analysis of published cancer genomics datasets, such as The Cancer Genome Atlas (TCGA), shows that the PAM pathway is one of the most frequently altered oncogenic pathways in gynecologic cancers (Figure 1B). In EC, more than 90% of patients present with genetic alterations in one or more PAM pathway genes, with *PIK3CA* and *PTEN* alterations detected in 54% and 68% of cases (Figure 1B). In OC and CC, *PIK3CA* genetic alterations are the most common, with a prevalence of 22% and 39%, respectively. A study combining next-generation sequencing and immunohistochemistry confirmed the high prevalence of *PI3KCA* mutations (37%, 29%, and 8%), *PTEN* mutations (33%, 4%, and 3%) and PTEN protein loss (49%, 30%, and 21%) in EC, CC, and OC, respectively [8]. Dysregulation of the PAM pathway can also occur in the absence of canonical PAM pathway mutations, e.g., due to the dysregulation of interconnected pathways (e.g., RTKs), or epigenetic mechanisms [5,9,10].

The role of the PAM pathway as a driver of tumor development and progression in cancer is well-established. The increased activation of PAM signaling in cancer cells affects multiple cellular functions; e.g., it promotes cell cycle progression, counteracts pro-apoptotic signals, and induces metabolic adaptations required to sustain tumor growth [5,6,7,11]. Since cancer cells heavily rely on these cellular functions, inhibitors targeting one or more nodes of the PAM pathway have been developed [1,3,5,12]. Several PAM inhibitors targeting a specific PAM pathway node (‘single-node’ PAM inhibitors) are FDA-approved for advanced breast cancer in combination with hormonal therapy; these inhibitors include everolimus (mTORC1 inhibitor), alpelisib (PI3Kα inhibitor), and capivasertib (AKT inhibitor). Currently, the combination of everolimus and letrozole, an aromatase inhibitor, is recommended by the National Comprehensive Cancer Network (NCCN) for patients with recurrent or metastatic endometrioid EC.

As a monotherapy, single-node PAM inhibitors have shown limited therapeutic efficacy or durability in solid tumors. The inhibition of single PAM pathway nodes can lead to drug resistance through intra-pathway feedback loops. Examples include increased PI3Kβ activity upon PI3Kα inhibition [13], the IRS1/2-mediated reactivation of PI3K-AKT signaling upon mTORC1 inhibition [14], and PAM signaling rebound through the inhibition of PTEN translation [15]. Moreover, when the PAM pathway is inhibited, compensatory pathways interconnected with the PAM pathway, such as the estrogen receptor (ER) pathway, the mitogen-activated protein kinases (MAPK) pathway, and the cyclin-dependent kinases (CDK) pathway, can drive tumor progression [3,16]. A comprehensive inhibition of multiple PAM pathway nodes, as well as the targeting of PAM-interconnected pathways, could minimize the resistance to PAM inhibitors and increase their therapeutic efficacy [3,17].

Perhaps one of the greatest limiting factors to therapeutic interventions of the PAM pathway is where PAM inhibitors, alone or in combination with other therapies, induce hyperglycemia and significant increases in insulin secretion. This not only represents a relevant clinical adverse event in patients (especially in patients with insulin deficiency or resistance) but can also lead to the reactivation of the PAM pathway in cancer cells, thus reducing drug efficacy [18,19,20,21]. An optimal multi-node PAM inhibitor would be one that effectively exerts anti-proliferative/cytotoxic effects in cancer cells while limiting adverse effects like hyperglycemia and hyperinsulinemia.

Gedatolisib is a pan-PI3K/mTOR inhibitor targeting all class I PI3K isoforms, mTORC1, and mTORC2 with similar potencies [22,23]. Non-clinical studies have shown that gedatolisib exerts potent growth-inhibitory effects in vitro and induces tumor growth inhibition in multiple xenograft cancer models, including EC and OC [22,23,24,25]. Initial clinical trials have further shown preliminary gedatolisib efficacy in various solid tumors, including gynecologic cancers [26,27,28,29,30]. For instance, a Phase 1 trial showed the preliminary efficacy of gedatolisib combined with carboplatin and paclitaxel in clear cell ovarian cancer [26], while a Phase 2 clinical trial in recurrent EC showed that gedatolisib met the clinical benefit response criteria in the stathmin-low patient subpopulation [28]. Gedatolisib also showed fewer class-associated adverse effects, such as hyperglycemia and gastrointestinal and skin toxicities, when compared to the published data for other PAM inhibitors [26,27,28,29,30,31,32,33]. Based on these encouraging results, a Phase 3 clinical trial (VIKTORIA-1, NCT05501886) was initiated to evaluate gedatolisib in combination with fulvestrant, with and without palbociclib, in patients with HR+/HER2− advanced breast cancer.

The present study investigated gedatolisib efficacy either as a single agent or in combination with other targeted therapies in multiple gynecologic cancer cell models. We first demonstrated that gedatolisib exerted more potent and efficacious anti-proliferative and cytotoxic effects than single-node PAM inhibitors (alpelisib, capivasertib, and everolimus) in EC, OC, and CC cell lines, regardless of the PAM pathway mutational status. We further demonstrated that the combination of gedatolisib with a CDK4/6 inhibitor (palbociclib) or an ER antagonist (fulvestrant) increased the gedatolisib in vivo efficacy in the OC and EC xenograft models, respectively. These results suggest that the multi-node inhibition of the PAM pathway by gedatolisib, in combination with anti-estrogens and/or CDK4/6 inhibitors, could be an effective therapeutic strategy for the treatment of gynecologic cancers.

## 2. Materials and Methods

### 2.1. Genomic Profiling of PAM Pathway Clinical Mutations

Genetic alterations in key PAM pathways genes (*PIK3CA*, *PIK3CB*, *PIK3R1*, *PIK3R2*, *PTEN*, *AKT1*, *AKT2*, and *AKT3*) were identified by cBioPortal (https://www.cbioportal.org/) analysis of the following public databases: uterine corpus endometrial carcinoma (TCGA, panCancer Atlas, n = 509 samples/patients analyzed), ovarian serous cystadenocarcinoma (TCGA, panCancer Atlas, n = 398 samples/patients analyzed), and cervical squamous carcinoma (TCGA, panCancer Atlas, n = 278 samples/patients analyzed). Mutations, structural variants, and putative copy number alterations were selected for analysis in the genomic profiles.

### 2.2. Cell Culture

Ovarian cancer, endometrial cancer, and cervical cancer cell lines were obtained from ATCC (Manassas, VA, USA), AcceGen Biotechnology (Fairfield, NJ, USA), Sigma-Aldrich (St. Louis, MO, USA), Sekisui XenoTech (Kansas City, KS, USA), DCTD Tumor Repository (Bethesda, MD, USA), and JCRB (Richmond, VA, USA)as listed in Table 1. Cells were authenticated by STR profiling (ATCC) and tested for mycoplasma. Cells were maintained based on the vendor’s recommendations in a 5% CO_2_ humidified incubator at 37 °C. Cells were passaged when sub-confluent and used for experiments within 2–3 passages. Driver alterations in key PAM pathway genes were identified by analysis of the Cancer Cell Line Encyclopedia (CCLE, Broad 2019 dataset) [34] through cBioPortal (https://www.cbioportal.org/). In the present study, the absence of driver alterations is referred to as ‘wild-type’ (wt). De-identified OC tumor tissue samples were used to establish OC primary cultures based on methods described in [35]. Liberty IRB (Columbia, MD, USA) granted IRB exemption because the research was determined not to involve human subjects per 45 CFR 46.102(f).

### 2.3. Treatments with PAM Inhibitors

A list of the PAM inhibitors used in this study is provided in Table 2. Gedatolisib, alpelisib, inavolisib, capivasertib, everolimus, copanlisib, dactolisib, and samotolisib used for in vitro treatments were obtained from Selleckchem (Houston, TX, USA). Drugs were reconstituted in DMSO and stored in aliquots at −80 °C for long-term storage, or at −30 °C for short-term storage before cell treatments. For cell viability, GR (growth rate inhibition) metrics, and flow cytometry assays, cells were seeded in two or more replicate wells on white 96-well plates coated with a mixture of collagen 1 (Advance Biomatrix, Carlsbad, CA, USA), fibronectin (Sigma-Aldrich), and laminin 332 (BioLamina, Sundbyberg, Sweden) in 180 µL culture medium and allowed to attach overnight. Preliminary experiments were conducted to identify cell-line-specific seeding densities that ensured that untreated cells did not reach full confluency by the end of the assay. During these preliminary experiments, cell viability was measured by RTGlo MT assay (Promega, Madison, WI, USA) at different time points to determine the doubling time of each cell line. After attachment, cells were treated with PAM inhibitors for the indicated time by adding 20 µL of 10× drug freshly diluted in medium. The final media volume after treatment was 200 µL. As a control, cells were treated with DMSO in the same amount used for drug treatments.

### 2.4. Cell Viability Assay

Cells were analyzed for cell viability at the end of a 72 h treatment with PAM inhibitors or DMSO by using the RT-Glo MT luciferase assay (Promega) as previously described [36]. A solution of RTGlo MT enzyme and substrate (both diluted 1:600) was prepared in warm medium, and 40 µL/well were added to the previously treated 96-well plates. After 1–1.5 h incubation in a cell culture incubator at 37 °C and 5% CO_2_, the RTGlo MT luminescence (live cells) was measured using an Infinite M1000 (Tecan) microplate reader. Wells with culture medium + RTGlo MT were used for background subtraction. After background subtraction, relative viability values were obtained by normalizing the relative light units (RLUs) to DMSO-treated cells (set as 1). PRISM 10.0.2 (GraphPad Software, Boston, MA, USA) was used to plot dose response curves (DRCs) and calculate absolute IC50 values. Cells were considered sensitive to gedatolisib if gedatolisib IC50 was <100 nM. Sensitivity cutoffs for alpelisib (3000 nM), capivasertib (3000 nM), and everolimus (50 nM) were based on previously published studies [37,38,39].

### 2.5. Proliferation-Normalized Inhibition of Growth Rate (GR) Assays

The normalized GR inhibition was calculated from RTGlo MT measurements before and after a 72 h treatment as described [40]. The normalized GR inhibition is calculated by using the formula GR(c,t) = 2k(c,t)/k(0) − 1 where GR(c,t) is the GR value for a drug at concentration “c” at time “t”, k(c,t) is the growth rate of drug-treated cells, and k(0) is the growth rate of untreated control cells. GR values and GR metrics were calculated with the online GR calculator tool [41] using previously calculated cell lines’ doubling times. Anti-proliferative effects are indicated by GR values between 0 and 1; cytotoxic effects are indicated by GR values between −1 and 0; and complete cytostasis is indicated by a GR value = 0. GR_50_ (concentration required to obtain a GR value = 0.5), GR_Max_ (GR value at the maximal concentration tested), and GR_AOC_ (area over the curve) were also calculated with the online GR calculator tool. The GR_50_ is a measure of drug potency, and the GR_Max_ is a measure of drug efficacy, while the GR_AOC_ captures variations in potency and efficacy at the same time without the constraint of curve fitting. PRISM was used to plot drugs DRCs.

### 2.6. Flow Cytometry

Cells treated for 48 h with PAM inhibitors or DMSO were harvested from 96-well plates and analyzed by flow cytometry as previously described [36]. During the last 2 h of treatment, cells were incubated with 10 µM 5-ethynyl-2′-deoxyuridine (EdU) (Thermo Fisher, Waltham, MA, USA). EdU is a nucleoside analog that is incorporated into newly synthesized DNA and is used to assess DNA replication. At the end of the treatment, both medium (potentially containing floating dead cells) and cells were collected. The medium was transferred to a deep-well 96-well plate, while the cells were washed with PBS (Corning, Corning, NY, USA) and detached by incubation with 0.25% Trypsin (Corning) + 0.5 mM EDTA (Amresco, Solon, OH, USA). Trypsin was blocked with 0.3% Ovomucoid trypsin inhibitor (Worthington), and cells were transferred to the same deep-well 96-well plate containing the medium collected previously. Plates were centrifuged at 300× *g* for 7 min at 4 °C, and the cell pellets were washed with PBS and stained for 15 min at room temperature with Zombie NIR viability dye (Biolegend, San Diego, CA, USA). After washing with PBS + 1% BSA, cells were fixed with 1.6% paraformaldehyde for 10 min at room temperature (Electron Microscopy Sciences, Hatfield, PA, USA) and permeabilized with cold ACS grade methanol (Sigma) for 15 min at 4 °C. After fixation and permeabilization, cells were sequentially assayed for EdU incorporation, phosphoRPS6, and phospho-4EBP1. EdU incorporation was detected by using the Click-iT EdU Alexa Fluor 647 kit (Thermo Fisher) per vendor’s instructions. After the Click-iT reaction, cells were washed with PBS + 1% BSA, and stained for 30 min at 4 °C with anti-p4EBP1-Alexa Fluor 488 (T36/T45) (BD Biosciences) diluted 1:25 and anti-pRPS6-BV421(S235/S236) (Biolegend) diluted 1:50. After washing with PBS + 1% BSA, samples were run on a Novocyte 3005 (Agilent, Santa Clara, CA, USA) flow cytometer. Data were analyzed by using NovoExpress 1.5.6 (Agilent). Cell debris was first excluded from the analysis by forward and side scatter gating. Subsequently, Zombie staining was used to gate the live cells, which were analyzed for pRPS6 and p4EBP1 levels (median fluorescence intensity after unstained background subtraction) as well as EdU incorporation (% of EdU+ cells). After normalization to DMSO-treated control cells (set at 1), data were analyzed in PRISM to obtain DRCs.

### 2.7. Drug Synergy Analysis

Drug synergy analysis was performed using the median effect principle proposed by Chou and Talalay [42]. The Calcusyn software (version 2.11) (https://norecopa.no/norina/calcusyn-version-20) was used to calculate the combination index (CI) and fraction affected (Fa). CI < 1 indicates synergism, CI = 1 indicates additivity, and CI > 1 indicates antagonism. The Fa value represents the effect of the drug on EdU incorporation, where Fa = 0 represents no inhibition and Fa = 1 represents 100% inhibition.

### 2.8. Quantitative PCR

Ishikawa cells were seeded on collagen1/fibronectin-coated 12-well plates at 4.5 × 10^4^ cells/well and left attached for approximately 24 h. Wells were then washed with serum-free medium, and medium was replaced with 1 mL of either standard growth medium (containing regular FBS) or E2-depleted medium (phenol-red free medium supplemented with charcoal-stripped FBS [R&D Systems]). After 24 h, cells were treated O/N (~16 h) with E2, gedatolisib, and fulvestrant, alone or in combination. For the combinations, E2 was added 15 min after addition of fulvestrant. After treatment, medium was removed, and RNA was extracted with QuickRNA Microprep kit (Zymo, Irvine, CA, USA) per manufacturer’s instructions. Up to 1 µg RNA was used for cDNA synthesis using the High-Capacity cDNA synthesis kit (Thermo Fisher). cDNA (40 ng/reaction) was used for qRT-PCR with TaqMan Fast Advanced Master Mix (Thermo Fisher) and Taqman probes for ESR1, PGR, GREB1, HPRT1, and ACTB (Thermo Fisher) on a QuantStudio 3 thermocycler (Thermo Fisher). Relative mRNA expression was calculated based on the ΔΔCt method [43] using both HPRT1 and ACTB as reference genes for normalization.

### 2.9. CELsignia PI3K Signaling Pathway Test

OC primary cultures at low passage were used for the CELsignia assay. After counting with a NucleoCounter NC-250 (Chemometec, Allerod, Denmark)), cells were transferred onto 96-well E-plates (Agilent) previously coated with a mixture of extracellular matrix proteins. An xCELLigence RTCA impedance biosensor (Agilent) was used to monitor real-time cell responses to 1-oleoyl lysophosphatidic acid (LPA) (Tocris, Minneapolis, MN, USA) with and without PAM inhibitors (gedatolisib, inavolisib) as described previously [35,36,44]. Following treatment with PAM inhibitors for 18 h, cells were stimulated with 125 nM LPA and monitored for impedance changes over an additional 4 h. TraceDrawer (Ridgeview Instruments AB, Uppsala, Sweden) was used for data analysis. The inhibition of the LPA signal by PAM inhibitors was calculated as described in [35].

### 2.10. Animal Studies

Ishikawa and SKOV3 xenograft studies were performed at Crown Bioscience, Inc. The protocols, procedures, and any amendment(s) used in the animal studies were reviewed and approved prior to execution by the Institutional Animal Care and Use Committee (IACUC) of CrownBio. Animal studies were compliant with the regulations of the Association for Assessment and Accreditation of Laboratory Animal Care (AAALAC). Female BALB/c nude mice (GemPharmatech, Nanjing, China) that are 6–8 weeks old were inoculated subcutaneously in the right upper region with SKOV3 (1 × 10^7^) or Ishikawa (1 × 10^7^) cells resuspended in 0.1 mL of PBS and Matrigel (1:1). Mice were kept in Polysulfone IVC cages under 12 h light/dark cycles in a humidity- and temperature-controlled environment. Mice had free access to sterilized food and water. When the mean tumor size reached approximately 100–200 mm^3^, mice were randomized for treatment based on the “Matched distribution” method (Study DirectorTM software, version 3.1.399.19). The date of grouping was denoted as day 0. Mice were treated with gedatolisib (Celcuity, Minneapolis, MN, USA), fulvestrant (Selleckchem), palbociclib (Selleckchem), or saline (vehicle control) as indicated. The in vivo dose of gedatolisib (15 mg/kg) was chosen based on our previous studies [36,45]. This concentration was lower than the equivalent human dose (180 mg/kg) used in patients. Gedatolisib was resuspended in H_2_O and administered intravenously (SKOV3 xenografts) or intraperitoneally (Ishikawa xenografts) every 4 days (Q4D); fulvestrant was resuspended in DMSO, mixed with corn oil, and administered subcutaneously Q4D; and palbociclib was resuspended in saline and administered orally daily (QD). The animals were checked daily for morbidity and mortality. The animals were routinely checked for behavioral changes such as food/water consumption and mobility, eye/hair matting, loss or gain of body weight, or other abnormalities. Body weights were measured twice per week. Tumor length and width were measured twice per week using a caliper, and the volume in mm^3^ was calculated using the formula, V = (L × W × W)/2, where V is tumor volume, L is tumor length, and W is tumor width. The tumor length was the longest tumor dimension, and the tumor width was the longest tumor dimension perpendicular to the length. Study Director TM (version 3.1.399.19) was used to measure body weights and tumor volumes. Animals were euthanized if they lost over 20% of their body weight relative to the weight at the first day of treatment. Mouse with tumor ulceration of approximately 25% or greater on the surface of the tumor were also euthanized.

### 2.11. RNA-Seq Data Processing

#### 2.11.1. RNA-Seq Data Retrieval for EnC Clinical Samples

We retrieved bulk-level RNA-seq count data and associated clinical metadata for 554 primary endometrial tumor samples available through the Genomic Data Commons (GDC) portal as part of the TCGA-UCEC project in the R statistical environment using the TCGAutils (version 1.20.4) and TCGAbiolinks (version 2.28.4) packages. Count data were available for a total of 60,600 transcripts and was filtered to retain only protein-coding transcripts (N = 19,962). The R package DESeq2 (version 1.38.2) was used to create a SummarizedExperiment object, and the data were further filtered to only retain genes with a count of at least 1 in at least 10 samples. The resulting count data for 19,341 genes were subjected to variance stabilizing transformation (vst) in the DESeq pipeline (blind set to FALSE). Detailed information on the samples used for our analysis is provided in Appendix A.

#### 2.11.2. Batch-Effect Correction and Principal Components Analysis (PCA)

Technical batch information for samples was obtained using the MBatch Omic browser tool and batch effects in the normalized count data were quantified using the DSC parameter implemented in MBatch R package (version 2.1.0, PCARegularStructures function, DSCPermutations set to 1000). To reduce any potential masking of biological signal due to batch effects, we used the removeBatchEffects procedure in the R package limma (version 3.54.2) using batch_id and tissue source site as batch variables and tumor grade as a variable of interest and confirming a consequent reduction in the DSC value using MBatch (DSC values of 0.37 vs. 0.13, Appendix A). The transformed, batch-corrected count data were assessed using PCA to verify the lack of batch-specific sample clustering and to obtain biologically relevant co-ordinates for tumor samples, which was used subsequently to infer a disease progression transcriptomic trajectory. We selected the top 2000 genes in the dataset with highest IQR for PCA analysis using the prcomp procedure from the R stats package (version 4.2.2). The first 5 PCs explained ~34.5% of the variance in the dataset (Appendix A). The results (see Section 3.6) clearly show positioning of tumor samples from endometrioid and serous cancer patients by histological type and tumor grade, irrespective of their batch identifiers (Appendix A).

#### 2.11.3. Trajectory Inference of Tumor Progression Pseudotime

A disease-progression-associated trajectory was inferred from the integrated set of clinical EnC tumor bulk RNA-seq using Slingshot (version 2.6.0), following dimensionality reduction and clustering. As inputs for Slingshot, we used PCA positioning of tumors (PC1–PC5) along with tumor grade as the clustering label and setting Grade 1 tumors as the starting seed cluster for inferring a trajectory (Appendix A). Slingshot first identifies a global MST (minimum spanning tree) structure that creates a “minimum distance” tree connecting all cluster centers. Next, Slingshot uses the MST global structure to fit simultaneous smooth principal curves to the dataset. These curves or trajectories are constructed such that distance from all sample points is minimized. For our dataset, Slingshot identified a single trajectory that showed a biologically valid progression from low- to high-grade tumor samples (see Section 3.6). Slingshot also assigns each sample a “pseudotime” value indicating its position along this trajectory with higher values corresponding to advanced disease progression. We could assign each sample a pseudotime value ranging from 0 to 170 (Appendix A).

#### 2.11.4. Correlation of Transcripts and Associated Pathways to Pseudotime

We leveraged the pseudotime value assigned to each sample to identify linear correlation of tumor pseudotimes with each of the 19,341 protein-coding genes using the Pearson’s correlation co-efficient implemented in the R stats package (cor.test function, version 4.3.0). In order to minimize the effect of outliers on the correlation analysis, we used a 20-times-repeated leave-one-third-out procedure by randomly drawing 20 sets of samples, each comprising two-thirds of the entire sample set, and then computing the median correlation co-efficient between pseudotime and individual mRNA levels across the 20 sample sets. Using this approach, we obtained a dataframe containing the median Pearson’s co-efficient and associated *p*-value for each of the protein-coding transcripts in the cohort (Appendix A). A ranked list of transcripts ordered by correlation co-efficient was subsequently used as input for gene set enrichment analysis using the R package fgsea (version 1.24.0).

#### 2.11.5. Survival Analysis

We assessed whether pseudotime-correlative transcripts are predictive of survival in patients with endometrial cancer using RNA-seq *FPKM-UQ* count data obtained from the TCGA-UCEC dataset (N = 547 patients, using the GDC portal) and carried out Kaplan–Meier analysis (2-group overall survival analysis, NULL values were removed) using the XenaPython package (https://github.com/ucscXena/xenaPython.git, accessed on 15 October 2024) for the following transcripts that were significantly (FDR < 0.05) either positively (*SRD5A1*, *AURKA*, *TRIB3*, *E2F1*, *TPX2*, and CCNE1) or negatively (*GREB1*, *MYB*, *TFF3*, *PGR*, *MLPH*, and ELOVL5) correlated with pseudotime (Appendix A).

### 2.12. Statistical Analyses

Statistical significance was calculated using PRISM (GraphPad) or Excel as indicated in the figure legends. Differences were considered significant when *p* < 0.05.

## 3. Results

### 3.1. Analysis of PAM Inhibitors’ Response in Gynecologic Cancer Cell Lines Using Growth Rate Metrics and Cell Viability Assays

The growth inhibitory effects of gedatolisib and other PAM inhibitors were evaluated in a panel of 24 ovarian cancer, endometrial cancer, and cervical cancer cell lines with various types of PAM pathway mutational status (Table 1). Cells were analyzed for cell viability by RTGloMT assay before and after a 72 h treatment with PAM inhibitors to assess endpoint cell viability as well as growth rate (GR) metrics. Differently from endpoint viability assays, the analysis of GR metrics is independent of cell doubling time (which can be a confounding factor when comparing cell lines with different growth rates) and allows the identification of both cytostatic and cytotoxic effects (Figure 2A) [40].

Based on the GR metrics analysis, gedatolisib exerted potent, dose-dependent anti-proliferative and cytotoxic effects (Figure 2B,C), with an average GR_50_ = 20 nM and GR_Max_ = −0.53 (Figure 2D). The potency (GR_50_) and efficacy (GR_Max_) of gedatolisib were similar in cell lines with or without alterations of key PAM pathway genes (e.g., *PIK3CA*, *PIK3R1*, *PTEN*, and *AKT)*, indicating that these metrics were not influenced by the PAM pathway mutational status (Figure 2D). This observation was further confirmed by the calculation of the GR_AOC_, which captures both efficacy and potency in the same metric (Figure 2E). Gedatolisib showed high potency and efficacy in all cancer types tested, with an average GR_AOC_ = 3.69, 2.58, and 2.76 in the OC, EC, and CC cancer cells lines, respectively (Appendix A).

The growth inhibitory effects of gedatolisib were then compared to PAM inhibitors targeting single nodes of the PAM pathway: alpelisib (PI3Kα), capivasertib (AKT), and everolimus (mTORC1) (see Table 2). In all cell lines tested, gedatolisib showed a greater average potency and efficacy than single-node PAM inhibitors (Figure 2D and Appendix A). The average gedatolisib GR_50_ (20 nM) was at least 50 times lower than the average GR_50_ of alpelisib (5414 nM), capivasertib (11783 nM), and everolimus (1126 nM) (Figure 2D). Gedatolisib exerted cytotoxic effects in all cell lines, with an average GR_Max_ = −0.53, while alpelisib, capivasertib, and everolimus exerted mostly anti-proliferative effects, with an average GR_Max_ = 0.40, 0.48, and 0.22, respectively (Figure 2D). In 12 cell lines, everolimus was more potent (lower IC_50_) than gedatolisib but did not reach the same level of efficacy (higher GR_Max_) (Figure 2D). The comparison of the GR_AOC_ values further demonstrated that gedatolisib was generally more potent and efficacious than the single-node PAM inhibitors regardless of the PAM pathway mutational status (Figure 2F). These results were also confirmed by an endpoint cell viability assay (Appendix A).

These experiments suggested that gedatolisib’s higher potency and efficacy relative to single-node PAM inhibitors was due to its ability to target multiple PAM pathway nodes. This hypothesis was supported by additional studies comparing gedatolisib with other multi-node PAM inhibitors, i.e., copanlisib, samotolisib (pan-PI3K and weaker mTOR inhibitors), and dactolisib (equipotent pan-PI3K/mTOR inhibitor) (see Table 2). Of these multi-node inhibitors, dactolisib was the only compound that showed a similar potency and efficacy to gedatolisib, with an average GR_AOC_ = 3.0 and 3.3, respectively (Appendix A). Samotolisib and copanlisib, which have weaker activity against mTOR relative to PI3K, were both less potent and efficacious relative to gedatolisib or dactolisib (average GR_AOC_ = 2.0 and 2.4 for samotolisib and copanlisib, respectively) (Appendix A). This suggests that, to achieve effective growth-inhibitory effects in gynecologic cancer cells, a PAM inhibitor should be able to target with equal, low nanomolar, potency both PI3K and mTOR.

Based on these results, we set out to test the effects of gedatolisib and single-node PAM inhibitors on the PAM pathway activity in a subset of OC and EC cell lines.

### 3.2. Analysis of PAM Pathway Activity in Response to PAM Inhibitors

The effect of gedatolisib and single-node PAM inhibitors on the PAM pathway activity was evaluated by measuring the phosphorylation status of 4EBP1and RPS6, two effectors downstream of both PI3K-AKT-mTORC1 and mTORC2-AKT (Figure 1A). Representative data are shown for the EC cell line AN3CA in Figure 3A, while a summary of pRSP6 and p4EBP1 inhibition in five EC cell lines is shown in Figure 3B. Gedatolisib decreased the levels of pRPS6 and p4EBP1 in all cell lines with an average inhibition of 70% for pRPS6 and 48% for p4EBP1 at 333 nM. At the same concentration, capivasertib and everolimus induced 51–61% inhibition of pRPS6, while alpelisib had a more modest effect (15% inhibition) (Figure 3C). Remarkably, the single-node PAM inhibitors had a much lower effect on p4EBP1 relative to gedatolisib, inducing an average p4EBP1 inhibition < 20% (Figure 3C).

The inhibitory effects of gedatolisib on the PAM pathway were also confirmed in OC cell lines and primary cultures. In the OC cell lines A2780 and SKOV3, gedatolisib induced a dose-dependent decrease in pRPS6 (Appendix A). At 333 nM, gedatolisib inhibited pRPS6 by 34% and 69% in A2780 and SKOV3, respectively, while alpelisib induced 23% and 28% inhibition, respectively (Appendix A). Similarly, the impedance-based CELsignia test [35,44] showed that gedatolisib significantly inhibited the PAM pathway activity stimulated by a well-known oncogenic GPCR agonist (125 nM LPA) in four OC primary cultures and in the OC cell line OVCAR3 (Appendix A). In contrast, the PIK3α inhibitor inavolisib (GDC-0077) had no or very modest effects on the LPA-stimulated PAM pathway activity (Appendix A).

Based on these results, the PAM pathway activity appeared to be inhibited more effectively by gedatolisib than single-node inhibitors in both EC and OC cells. We hypothesized that the differential inhibition of the PAM pathway activity by gedatolisib and single-node PAM inhibitors could result in a differential effect on DNA replication and cell proliferation.

### 3.3. Analysis of DNA Replication in Response to PAM Inhibitors

The PAM pathway controls multiple cellular functions, including DNA replication and cell cycle [5,7]. To test whether gedatolisib and single-node PAM inhibitors had different effects on DNA replication, cells were incubated with the thymidine analog EdU during the last two hours of a 48 h treatment with PAM inhibitors. During the S phase of the cell cycle, EdU is incorporated into newly synthesized DNA and can be detected by flow cytometry as shown in Figure 4A. Consistent with the growth-inhibitory effects observed by GR metrics analysis, gedatolisib induced a potent and dose-dependent inhibition of EdU incorporation in EC cells (Figure 4B,C). On average, 333 nM gedatolisib inhibited EdU incorporation by 70%, while alpelisib, capivasertib, and everolimus induced <35% inhibition (Figure 4D). The Ishikawa cell line, which had the highest GR_50_ (158 nM) among the EC cell lines tested, was also the only EC cell line in which 333 nM gedatolisib induced <50% inhibition of EdU incorporation (Figure 4D).

Gedatolisib was also effective in OC cell lines and primary cultures. In the OC cell lines A2780 and SKOV3 (Appendix A), gedatolisib inhibited EdU incorporation by 71% and 97%, respectively, at 333 nM (Appendix A). At the same concentration, alpelisib induced ≤ 20% inhibition (Appendix A). Similarly, gedatolisib inhibited EdU incorporation more effectively than inavolisib (39–94% versus 2–10%, respectively) in four OC primary cultures and in the OVCAR3 cell line (Appendix A).

These results demonstrated that gedatolisib blocked DNA replication more effectively than the single-node PAM inhibitors in EC and OC cells.

### 3.4. Combination of Gedatolisib and Palbociclib in the SKOV3 Ovarian Cancer Model

Several non-clinical and clinical studies support the combination of PAM inhibitors and CDK4/6 inhibitors in tumors other than breast cancer, based on the rationale that these two pathways can compensate for each other when these inhibitors are used as single agents [46,47,48].

Non-clinical studies suggest that the benefit of the concomitant targeting of PAM and CDK pathways mostly relies on the exacerbation of cell cycle inhibition [48,49]. We used the SKOV3 OC cell model to test the effects of the gedatolisib/palbociclib combination on DNA replication. As shown in Figure 5A, gedatolisib (10–30 nM) and palbociclib (100 nM) reduced EdU incorporation by 35–70% and 40%, respectively, as single drugs. The combination of gedatolisib and palbociclib reduced EdU incorporation significantly more than the single drugs and almost completely blocked DNA replication at the highest concentrations tested (Figure 5A). An additional analysis by the Chou–Talalay method indicated that gedatolisib and palbociclib synergistically inhibited DNA replication (CI values < 1 for all combinations tested) (Figure 5B).

The gedatolisib/palbociclib combination was further tested in vivo. Nude mice bearing SKOV3 xenograft tumors were treated with vehicle, gedatolisib (15 mg/kg), or a combination of gedatolisib (15 mg/kg) and palbociclib (50 mg/kg) for 28 days. As a single agent, gedatolisib induced 68% TGI at the end of the treatment (Figure 5C, left). The combination of palbociclib and gedatolisib reduced tumor growth significantly more than gedatolisib alone, inducing a TGI of 92% (Figure 5C, left). The mice body weight did not show significant changes during treatment, indicating a lack of toxicity (Figure 5C, right).

These non-clinical studies confirmed that gedatolisib exerted tumor growth inhibition as a single agent in an OC xenograft model and indicated that gedatolisib growth-inhibitory effects can be significantly increased by the combination with palbociclib.

These results support the evaluation of gedatolisib in combination with the CDK4/6 inhibitor palbociclib in patients with solid tumors in an ongoing clinical trial (NCT03065062).

### 3.5. Combination of Gedatolisib and Fulvestrant in the Ishikawa Endometrial Cancer Model

Due to the crosstalk between the PAM and estrogen signaling pathways, the inhibition of either signaling pathway can lead to the activation of the other [50,51]. To address these resistance mechanisms, several PAM inhibitors, in combination with hormone therapy, are approved to treat HR+/HER2− advanced breast cancer. Additionally, clinical trials treating patients with endometrial cancer with everolimus in combination with hormonal therapy have reported encouraging results [52,53].

PI3K inhibition by alpelisib has been previously demonstrated to increase ERα transcriptional activity in MCF7 breast cancer cells. As a consequence of the MCF7 enhanced dependency on estrogen, the addition of the anti-estrogen fulvestrant to alpelisib increased the single agent alpelisib growth-inhibitory action [50]. We first tested whether gedatolisib similarly affected the ER signaling in the endocrine-sensitive Ishikawa EC model [54]. Gedatolisib (100 nM) increased the mRNA levels of two ERα-target genes, *PGR* (Figure 6A) and *GREB1* (Figure 6B), indicating the activation of ERα transcriptional activity. The increase in *PGR* and *GREB1* levels was observed both in the growth medium (containing picomolar concentrations of E2) and in the E2-depleted medium (i.e., medium supplemented with charcoal-stripped FBS), especially after supplementation with E2. As shown in Figure 6B, the treatment with 1 nM E2 promoted the transcription of both *PGR* and *GREB1* mRNAs, and the addition of 100 nM gedatolisib further increased their levels by approximately 30–50%. The increase in *PGR* and *GREB1* mRNAs was effectively inhibited by the addition of 100 nM fulvestrant, a selective estrogen receptor degrader (SERD) (Figure 6A,B).

After confirming that the Ishikawa cell line was a suitable EC model for the estrogen response, we tested whether the combination of gedatolisib and fulvestrant affected the Ishikawa tumor growth in vivo. Ishikawa xenografts in nude mice were treated with gedatolisib and fulvestrant, alone or in combination, at low or high doses. At low doses, gedatolisib (7.5 mg/kg) and fulvestrant (60 mg/kg) administered as single agents induced only modest (<30%) and not statistically significant TGI. However, the combination of gedatolisib (7.5 mg/kg) and fulvestrant (60 mg/kg) increased gedatolisib TGI from 20% to 56% (*p* < 0.01) (Figure 6C, left). At higher doses, single-agent gedatolisib (15 mg/kg) or fulvestrant (120 mg/kg) induced 46–48% TGI, which was statistically significant relative to vehicle-treated tumors. The combination of gedatolisib (15 mg/kg) and fulvestrant (120 mg/kg) further increased the TGI (64%) (Figure 6C, right). The body weight of the study mice did not show significant changes during treatment, indicating a lack of toxicity (Figure 6D).

Overall, the experiments with the Ishikawa cell line showed that gedatolisib can increase estrogen signaling, and that the inhibition of estrogen signaling with fulvestrant can increase the gedatolisib in vivo efficacy in an EC cell model.

### 3.6. Transcriptomic Trajectory Analysis of EC Clinical Tumor Progression

Alterations in the DNA sequence of key EC driver genes, including components of the PAM and its co-regulatory pathways, are binary (i.e., mutated or non-mutated) and often occur concomitantly, complicating a precise understanding of transformed tumor functions during disease progression. Several recent studies have leveraged the transcriptome-wide gene expression profiles of clinical tumor specimens to obtain a more complete and quantitative measure of cancer progression [55,56,57,58,59]. To understand the relevance of our non-clinical results reported above and gather further insight into the functional pathways dysregulated along with PAM signaling in human EC tumors, we applied trajectory inference, a method that allows the pseudo-temporal reconstruction of biological specimens clustered by their whole transcriptome states [55], to characterize the EC tumor progression in patients. The approach first clustered 554 individual EC tumor specimens from the TCGA-UCEC cohort based on their RNA-seq transcriptomic profiles (Figure 7A,B). The Slingshot analysis (see Section 2) defined a single lineage in three-dimensional PC space (i.e., an ordered set of tumor samples) that correlated biologically with the tumor grade (Figure 7C), and, within this lineage, assigned tumor pseudotimes (i.e., a one-dimensional variable representing each tumor’s transcriptional progression toward the terminal state) (Appendix A).

Using this model, we visualized gene expression changes along the trajectory (Figure 7D) and identified gene sets and pathways enriched along the path to EC progression. Among the most enriched genes, we identified key members of PAM signaling, the cell cycle (E2F response, mitotic spindle), the estrogen response, and the interferon response (Figure 7E,F). The progression path demonstrates with high confidence (Pearsons *p* < 0.01, Appendix A) that EC evolution involves combined alterations of multiple pathways, including the PAM pathway, the cell cycle, the estrogen pathway, and the fatty acid metabolism and glycolysis pathways (Appendix A). These transcriptional changes were associated with different patient survival outcomes as shown by results from the Kaplan–Meier survival analysis of EC patients for transcripts selected from the pseudotime-inferred trajectory (Appendix A).

Collectively, the whole transcriptome analysis of EC patients demonstrates that EC progression is associated with alterations in the PAM pathway as well as alterations in other pathways, which support the rationale for combining PAM inhibitors like gedatolisib together with approved therapies targeting estrogen signaling (e.g., ER inhibitors) and/or cell cycle progression (e.g., CDK inhibitors) in the clinic.

## 4. Discussion

The key role of the PAM pathway in controlling the cellular and metabolic functions required for cancer cell survival and proliferation and its frequent dysregulation in cancer cells have made this pathway an attractive therapeutic target in many cancer types, including gynecologic cancers. However, many PAM inhibitors showed relatively low clinical efficacy as monotherapy. Single-node PAM inhibitors can induce resistance or adaptive mechanisms associated with feedback loops, feed forward loops, or compensatory mechanisms occurring when only one node of the PAM pathway is blocked. Reactivated PAM leads to increased PIP3, which participates in regulating many cell functions especially through small GTPases. In addition, the inhibition of the PAM pathway can be counteracted by the activation of other interconnected pathways, such as the ER and CDK pathways. These limitations could be potentially overcome by (1) a more comprehensive inhibition of multiple nodes of the PAM pathway such as the pan-PI3K/mTOR inhibitor gedatolisib and (2) a combination of PAM inhibitors with drugs targeting PAM-interconnected pathways. The non-clinical data presented here demonstrate that gedatolisib, an equipotent pan-PI3K/mTOR inhibitor, exerts greater anti-proliferative and cytotoxic effects than single-node PAM inhibitors such as alpelisib (PI3Kα inhibitor), capivasertib (AKT inhibitor), and everolimus (mTORC1 inhibitor) in the EC, OC, and CC cell lines. The data further demonstrate that gedatolisib’s efficacy is enhanced by the combination with a CDK4/6 inhibitor (palbociclib) or an ER antagonist (fulvestrant) in the OC and EC xenograft models, respectively.

The potencies and efficacies of gedatolisib and other PAM inhibitors were compared by using two different approaches: classical cell viability assays based on an endpoint analysis of cell viability, and GR metrics analyses based on the assessment of cell viability before and after treatment. The two approaches showed that gedatolisib, by inhibiting multiple PAM pathway nodes, induced larger anti-proliferative/cytotoxic effects compared to inhibitors targeting single PAM pathway nodes. Of note, the greater effects of gedatolisib were observed regardless of the mutational status of key PAM pathway genes (e.g., *PIK3CA* and *PTEN*). These results are in line with a study by Weigelt et al. reporting that the dual PI3K/mTOR inhibitor PF-04691502, which has a similar potency and specificity as gedatolisib [60], reduced cell growth more effectively than GDC-0941 (PI3K inhibitor) or temsirolimus (mTOR inhibitor) in EC cell lines [61]. Moreover, our recent studies demonstrated that gedatolisib was more effective than alpelisib, capivasertib, and everolimus in both breast cancer cell lines [36] and prostate cancer cell lines [45], indicating that the greater efficacy of gedatolisib versus single-node PAM inhibitors can be generalized to multiple cancer types.

Although limited to a relatively small number of EC, OC, and CC cell lines, our study showed that the sensitivity to gedatolisib or the other PAM inhibitors tested was not significantly associated with mutations in key PAM pathway genes (e.g., *PIK3CA* and *PTEN*). The role of PAM pathway mutations in the response to PAM inhibitors in patients with gynecological cancers is still unclear. An early study across multiple clinical trials in advanced breast, cervical, endometrial, and ovarian cancers showed a higher response to PI3K/mTOR inhibitors in patients with *PIK3CA* mutations versus patients without mutations [62]. However, a more recent OC meta-analysis showed that biomarkers of PAM activity were not useful in predicting clinical benefit [63]. Other studies found no significant correlation between *PIK3CA* mutations or PTEN loss and the response to PI3K, AKT, or mTOR inhibitors in EC [64,65,66]. These seemingly contrasting observations might be reconciled by evidence that the PAM pathway can be aberrantly activated in cancer cells due to multiple genetic and non-genetic factors. For instance, a proteogenomic study of PAM pathway alterations in over 11,000 TCGA human cancer samples discovered a substantial fraction of cancers, including gynecologic cancers, with high PAM pathway activity and no canonical mutations of PAM pathway genes [10].

The greater potency and efficacy of gedatolisib relative to single-node PAM inhibitors is likely explained by gedatolisib’s greater inhibition of the PAM pathway activity and PAM-controlled cellular functions, such as cell proliferation. To monitor the PAM pathway activity, we evaluated the phosphorylation status of RPS6 and 4EBP1, two PAM pathway effectors frequently used in non-clinical and clinical studies [67]. Both pRPS6 and p4EBP1 showed a substantial and durable decrease in association with a strong inhibition of DNA replication. Single-node PAM inhibitors were generally less effective than gedatolisib at inhibiting p4EPB1, even when they reduced pRPS6 levels (e.g., capivasertib and everolimus). These results are consistent with the published evidence that both PI3K and mTORC1 inhibition are necessary to induce effective 4EBP1 dephosphorylation [68]. In addition, it has been reported that the suppression of 4EBP1 phosphorylation is necessary to achieve maximal cancer cell growth inhibition [69]. Gedatolisib’s greater efficacy in inhibiting cell proliferation could be due, at least in part, to its ability to inhibit 4EBP1 activity more durably and efficiently than single-node PAM inhibitors. One of the main functions of 4EBP1 is to regulate protein translation upon mTORC1 activation. Since cancer cells heavily rely on protein synthesis to sustain their increased proliferation rates, the effective inhibition of protein synthesis via mTORC1-4EBP1 is expected to severely impact cancer cell growth [70].

The bioinformatics analysis of the EC progression trajectory of patients with endometrial cancer presented here indicated that transcriptional alterations of PAM pathway genes were associated with transcriptional alterations of other pathways relevant for tumor initiation and progression. The pathways identified by this analysis included the estrogen pathway, pathways related to cell cycle and DNA replication, and metabolic pathways such as glycolysis and fatty acid metabolism. In agreement with these observations, we recently showed that gedatolisib affects multiple PAM-controlled functions in breast and prostate cancer cell models [36,45]. Besides blocking DNA replication, reducing protein synthesis, and promoting apoptosis, gedatolisib also inhibited critical metabolic functions, including glycolysis. Like other tumor types, gynecologic tumors undergo extensive metabolic reprogramming (most notably increased glycolysis and glucose uptake) that sustains cancer cells’ demand for energy and biomolecule synthesis, and many of these metabolic changes are dependent on the PAM pathway’s increased activation [6,71,72]. Gedatolisib, by inducing a more comprehensive inhibition of the PAM pathway, could also reduce cancer cells’ essential metabolic functions more effectively than single-node PAM inhibitors. In future studies, testing whether the gedatolisib efficacy in gynecological cancer cells is also associated with the inhibition of key metabolic functions controlled by the PAM pathway will be important.

Clinical studies [29,30] have shown that patients treated with gedatolisib have a low incidence of Grade 3–4 hyperglycemia and did not cause study discontinuation compared to the published data with other inhibitors such as alpelisib and everolimus [31,32]. Insulin signaling through the PAM pathway is well-established [20]. Single-node PAM inhibitors can induce significant increases in hyperglycemia and insulin secretion, which leads to the reactivation of the PAM pathway in patients and a reduction in single-node drug efficacy [20,21]. The biochemical signals between cells of different tissues such as the pancreas, liver, fat, and skeletal muscle by glucagon and insulin are likely candidates responsible for hyperglycemia and the loss of antagonism by the single-node inhibitors. Currently, there is no clear biochemical explanation for the significant differences in hyperglycemia for single-node inhibitors versus gedatolisib.

A comprehensive inhibition of multiple PAM pathway nodes by pan-PI3K/mTOR inhibitors like gedatolisib could increase efficacy by preventing resistance mechanisms associated with single-node PAM inhibition. However, the efficacy of any PAM inhibitor can be limited by compensatory signaling pathways, such as the ER and CDK pathways, interconnected with the PAM pathway. This provides a strong rationale for combining PAM inhibitors with anti-estrogens and/or CDK inhibitors in hormone-driven cancers, particularly breast cancer [73]. Clinical studies such as SOLAR-1 [74], CAPItello-291 [75], and BOLERO-2 [76] supported the FDA approval of PAM inhibitors (alpelisib, capivasertib, and everolimus) in combination with hormonal therapy for patients with ER+/HER2− breast cancer. Additionally, a Phase 1b study recently reported a promising objective response rate in patients with advanced breast cancer treated with gedatolisib plus palbociclib and endocrine therapy [29]. Similar combinations have been or are being tested in other cancer types, including gynecologic cancers. In a randomized Phase 2 trial, the combination of everolimus and letrozole demonstrated clinically meaningful efficacy in patients with recurrent EC [52]. Another Phase 2 study is currently evaluating everolimus and letrozole, with or without ribociclib, in patients with advanced or recurrent EC (NCT03008408). Our bioinformatics analysis of 554 EC tumor samples showing concomitant alterations of PAM, the cell cycle, and estrogen pathways in the disease progression trajectory lends further support to the combination of PAM inhibitors, hormone therapy, and cell cycle inhibitors in the treatment of EC.

The present study demonstrates that the in vivo efficacy of gedatolisib was significantly increased by the combination with either an ER antagonist (fulvestrant) or a CDK4/6 inhibitor (palbociclib) in EC and OC xenograft models, respectively. From a mechanist standpoint, gedatolisib increased the ERα transcriptional function in Ishikawa EC cells, as previously observed in MCF7 breast cancer cells treated with the PI3K inhibitor alpelisib [50]. While the increase in ERα signaling can be considered an adaptive resistance mechanism, it can also enhance cancer cells’ dependency on estrogen and make them more vulnerable to the combined treatment with PAM inhibitors and anti-estrogens [50]. In the SKOV3 OC model, the combination of gedatolisib and palbociclib exerted a synergistic inhibitory effect on DNA replication in vitro and induced a greater TGI than gedatolisib alone in xenograft studies. Palbociclib, by blocking the CDK4/6-cyclin D-RB axis, prevents the progression through the G1 phase. However, cancer cells can rapidly adapt to palbociclib by restoring the cell cycle progression and DNA replication [46]. The combination of gedatolisib with ER antagonists and/or CDK4/6 inhibitors could counteract these early adaptations, resulting in a more durable and effective cell cycle blockade.

## 5. Conclusions

The frequent reliance of tumors on the dysregulation of the PAM pathway makes this pathway an attractive therapeutic target in gynecologic cancers. The NCCN Guideline recommending treatment with everolimus and endocrine therapy for some patient sub-groups with EC is partly reflective of this rationale. The present non-clinical study shows that gedatolisib, a pan-PI3K/mTOR inhibitor, exerts greater anti-proliferative and cytotoxic effects than single-node PAM inhibitors such as everolimus in the EC, OC, and CC cancer cell lines, regardless of their PAM pathway mutational status. In addition, the in vivo efficacy of gedatolisib was enhanced by the addition of fulvestrant or palbociclib, thus providing a strong rationale for evaluating gedatolisib in combination with endocrine therapy and/or CDK4/6 inhibitors in patients with OC or EC.

## Figures and Tables

**Figure 1 cancers-16-03520-f001:**
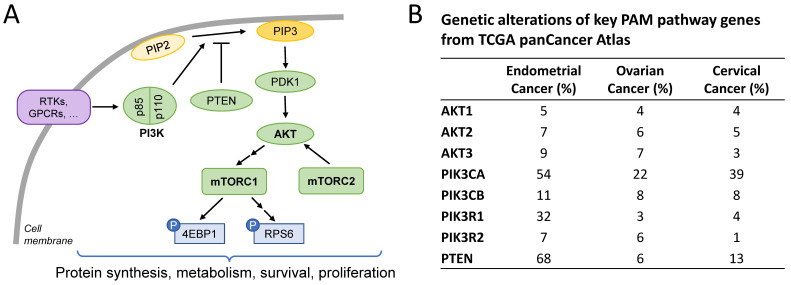
The PAM pathway is frequently dysregulated in gynecologic cancers. (**A**) Simplified scheme showing main pathway nodes (bold). (**B**) cBioPortal analysis of the TCGA panCancer Atlas showing the percentage of genetic alterations of key PAM pathway genes in endometrial cancer (509 samples analyzed), ovarian cancer (398 samples analyzed), and cervical carcinoma (278 samples analyzed).

**Figure 2 cancers-16-03520-f002:**
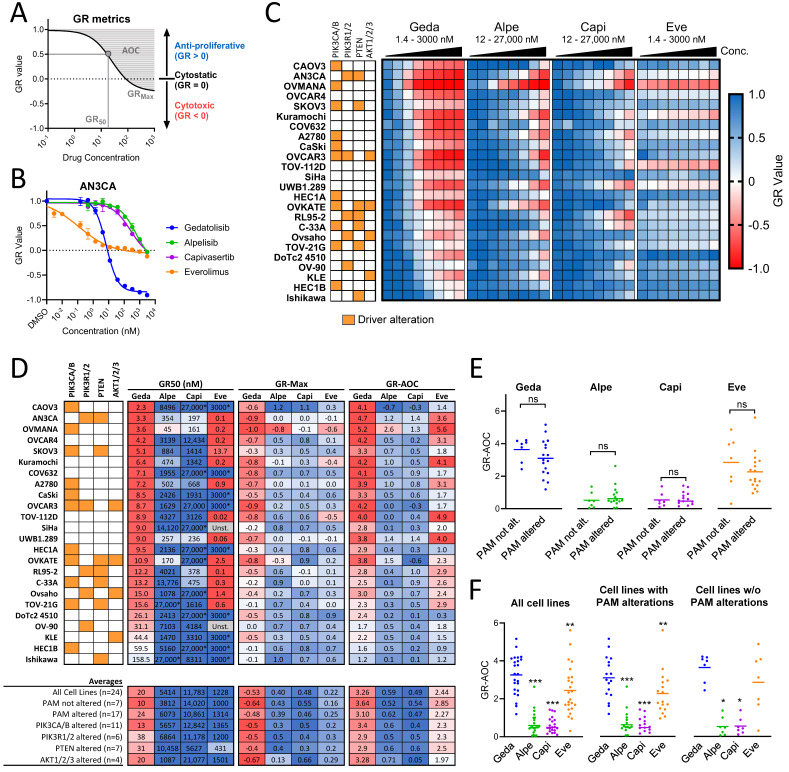
Analysis of PAM inhibitors response in gynecologic cancer cell lines using growth rate metrics. (**A**) GR metrics can be used to assess drugs’ anti-proliferative effects (GR value = 0–1), cytotoxic effects (GR < 0), potency (GR_50_), and efficacy (GR_Max_). Efficacy and potency can be also captured at the same time by calculating the area over the curve (GR_AOC_). Lower GR_50_ indicates higher potency; lower GR_Max_ indicates higher efficacy; and higher GR_AOC_ indicates higher potency and efficacy. (**B**) AN3CA GR values calculated by RTGlo MT assay before and after a 72 h treatment with PAM inhibitors are shown as an example. Data represent mean ± SD (n = 2 biologically independent samples). (**C**) Heatmap showing GR values in 24 gynecologic cancer cell lines treated with increasing concentrations of PAM inhibitors for 72 h. Concentrations shown in the heatmap = 1.4, 4.1, 12, 37, 111, 333, 1000, 3000, 9000, and 27,000 nM. See Appendix A for data. (**D**) GR_50_, GR_Max_, and GR_AOC_ values for gedatolisib and single-node PAM inhibitors in gynecologic cancer cell lines. Average values in subpopulations with or without altered PAM pathway genes are shown. * = max concentration tested, GR_50_ not reached; Unst. = unstable due to poor curve fitting that prevented reliable GR_50_ calculation. (**E**,**F**) GR_AOC_ analysis comparing potency and efficacy of gedatolisib and single-node PAM inhibitors in cell lines with or without driver genetic alterations in key PAM pathway genes (*PIK3CA/B*, *PIK3R1/2*, *PTEN*, and *AKT1/2/3*); ns = not significant, * = *p* < 0.05, ** = *p* < 0.01, and *** = *p* < 0.001 by Mann–Whitney test (top) or Wilcoxon matched-pairs signed-rank test (bottom). Significance in panel (**E**) is relative to cell lines without PAM alterations; significance in panel (**F**) is relative to cell lines treated with gedatolisib.

**Figure 3 cancers-16-03520-f003:**
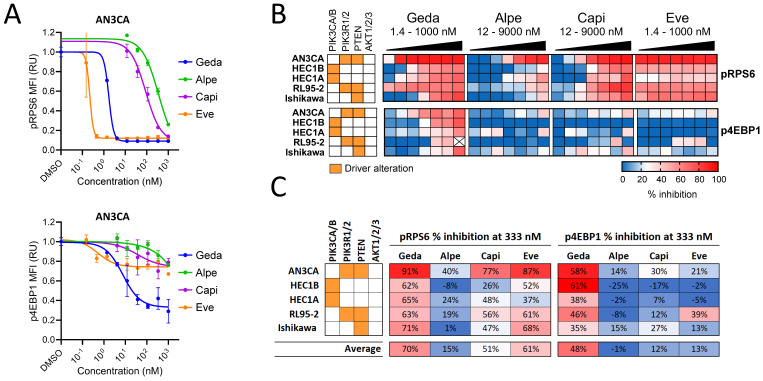
Analysis of PAM pathway activity in response to PAM inhibitors in EC cell lines. (**A**) PAM pathway activity in response to a 48 h treatment with PAM inhibitors was assessed by flow cytometry analysis of pRPS6(S235/S236) and p4EBP1(T36/T45) levels. The median fluorescence intensity (MFI) was normalized to DMSO-treated cells (set as 1) and used to plot PAM inhibitors DRCs as shown here for AN3CA as an example. Concentrations shown in the heatmap = 1.4, 4.1, 12, 37, 111, 333, 1000, 3000, and 9000 nM. Data represent mean ± SD (n = 2 biologically independent samples). (**B**) Heatmaps showing pRPS6 and p4EBP1 levels in five EC cell lines treated with increasing concentrations of PAM inhibitors. The % inhibition is relative to DMSO-treated cells. See Appendix A for data. (**C**) % inhibition of pRPS6 and p4EBP1 in response to 333 nM PAM inhibitors shows that gedatolisib is, on average, more efficacious than single-node PAM inhibitors.

**Figure 4 cancers-16-03520-f004:**
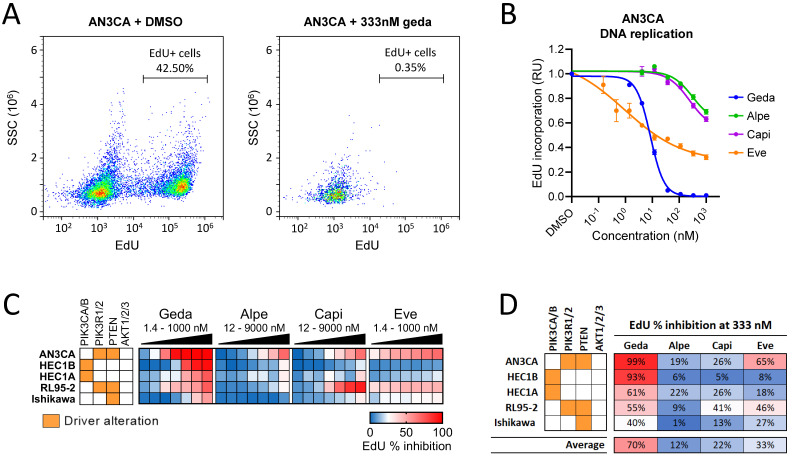
Analysis of DNA replication in response to PAM inhibitors in EC cell lines. (**A**) Example of EdU incorporation analysis by flow cytometry in AN3CA treated with 333 nM gedatolisib or DMSO (vehicle control) for 48 h. (**B**) The % of EdU+ cells was normalized to DMSO-treated cells (set as 1) and used to plot PAM inhibitors DRCs as shown here for AN3CA as an example. Data represent mean ± SD (n = 2 biologically independent samples). (**C**) Heatmap showing inhibition of EdU incorporation in a panel of five EC cell lines response treated with increasing concentrations of PAM inhibitors for 48 h. Concentrations shown in the heatmap = 1.4, 4.1, 12, 37, 111, 333, 1000, 3000, and 9000 nM. See Appendix A for data. (**D**) % inhibition of EdU incorporation in response to 333 nM PAM inhibitors shows that gedatolisib is, on average, more efficacious than single-node PAM inhibitors.

**Figure 5 cancers-16-03520-f005:**
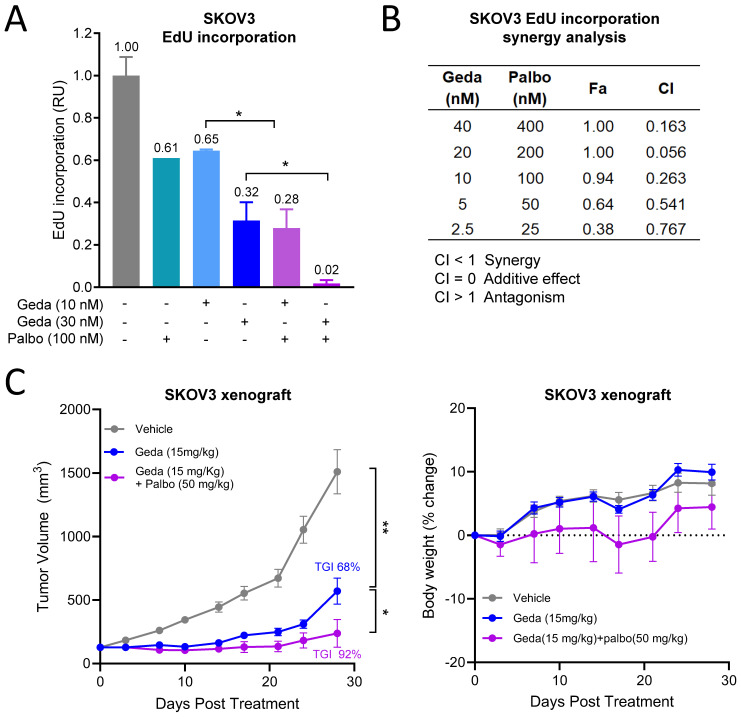
Combination of gedatolisib and palbociclib in the SKOV3 ovarian cancer model. (**A**) Flow cytometry analysis showing that the combination of gedatolisib and palbociclib inhibits EdU incorporation significantly more than the single drugs in SKOV3 cells. Data represent mean ± SD (n = 2 biologically independent samples). * *p* < 0.01 by unpaired two-sided *t*-test. (**B**) Chou–Talalay analysis showing that gedatolisib and palbociclib inhibit EdU incorporation synergistically. CI = combination index (values < 1 indicate synergy); Fa = fraction affected (0 indicates no inhibition and 1 indicates 100% inhibition). (**C**) SKOV3 cells were inoculated subcutaneously in the flank of BALB/c nude mice, and animals were treated with vehicle, gedatolisib (i.v. Q4D), or palbociclib (p.o. QD). Tumor growth curved and tumor growth inhibition (TGI) at the indicate time point is shown on the left. Mice body weight change during treatment is shown on the right. Data represent mean tumor volume ± SEM (n = 9–10 mice/arm). Statistical significance was calculated by one-way ANOVA; * *p* < 0.05, ** *p* < 0.01.

**Figure 6 cancers-16-03520-f006:**
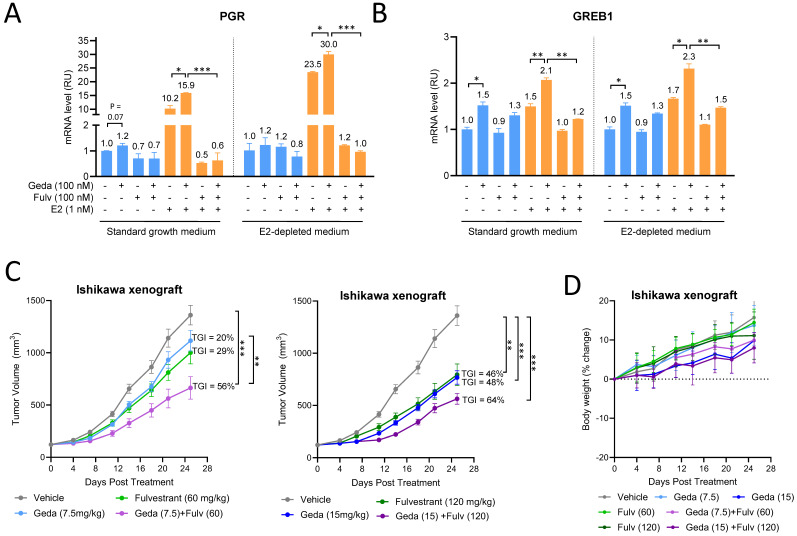
Combination of gedatolisib and fulvestrant in endometrial cancer models. (**A**,**B**) qPCR analysis showing that gedatolisib increased the transcription of two ERα-target genes, *PGR* (**A**) and *GREB1* (**B**), under different growth media conditions. * *p* < 0.05, ** *p* < 0.01, and *** *p* < 0.001 by unpaired two-sided *t*-test. (**C**) Ishikawa cells were inoculated subcutaneously in the flank of BALB/c nude mice, and animals (n = 10/arm) were treated with vehicle, gedatolisib (i.p. Q4D), fulvestrant (s.c. Q4D), or gedatolisib + fulvestrant at the indicated doses. Tumor growth inhibition (TGI) was calculated from tumor volumes at day 25 from beginning of the treatment. Data represent mean tumor volume ± SEM (n = 10 mice/arm). Statistical significance was calculated by one-way ANOVA; ** *p* < 0.01, *** *p* < 0.001. (**D**) Assessment of mice body weights during treatment.

**Figure 7 cancers-16-03520-f007:**
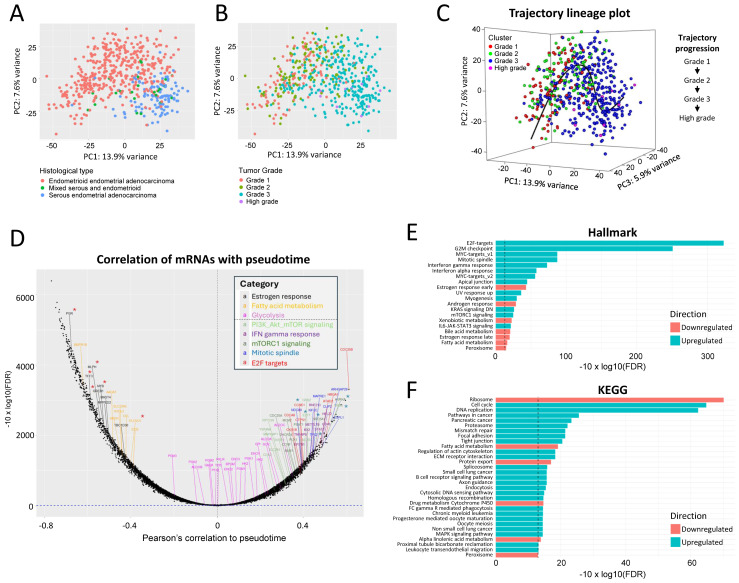
Trajectory of endometrial cancer progression. (**A**,**B**) Plots of EC tumor transcriptomes obtained from TCGA (N = 554 patients × 19,341 filtered transcripts) following dimensionality reduction by principal components analysis (PCA), colored either by tumor histological type (**A**) or by tumor grade (**B**). (**C**) Unbiased trajectory analysis of EC disease progression based on clinical tumor transcriptomic states. The inferred lineage of samples categorized by tumor grade is shown in the box below. (**D**) Plot of correlation between mRNAs and pseudotime inferred along the EC disease progression trajectory. Selected transcripts related to different hallmark pathway gene sets are shown. The horizontal dotted line indicates a cutoff value of FDR = 0.05. The transcripts marked with red and blue asterisks were chosen for further analysis (shown in Appendix A). (**E**,**F**) Gene set enrichment analysis of transcripts ranked for their correlation to pseudotime. The vertical dotted line indicates a cutoff value of FDR = 0.05.

**Table 1 cancers-16-03520-t001:** Cell lines used in this study.

Cell Line	Source	Cancer Type	PIK3CA	PIK3CB	PIK3R1	PIK3R2	PTEN	AKT1	AKT2	AKT3
AN3CA	ATCC	EC	-	-	Mut	-	Mut	-	-	-
HEC1A	ATCC	EC	Mut	-	-	-	-	-	-	-
HEC1B	ATCC	EC	Mut, AMP	AMP	-	-	-	-	-	-
ISHIKAWA ^1^	AcceGen Biotech.	EC	-	-	-	-	Mut	-	-	-
KLE	ATCC	EC	-	-	-	-	-	-	AMP	-
RL952	ATCC	EC	-	-	Mut	-	Mut	-	-	-
C33A	ATCC	CC	Mut	-	-	-	Mut	-	-	-
CASKI	ATCC	CC	Mut	-	-	-	-	-	-	-
DOTC24510	ATCC	CC	-	-	-	-	-	-	-	-
SIHA	ATCC	CC	-	-	-	-	-	-	-	-
A2780	Sigma-Aldrich	OC	Mut	-	-	-	-	-	-	-
CAOV3	ATCC	OC	Mut	AMP	-	-	-	-	-	-
COV362	Sigma-Aldrich	OC	-	-	-	-	-	-	-	-
KURAMOCHI	Sekisui XenoTech	OC	-	-	-	-	-	-	-	-
OV90	ATCC	OC	-	-	-	HOMDEL	-	-	-	-
OVCAR3	ATCC	OC	-	AMP	Mut	-	-	-	AMP	-
OVCAR4	DCTD Tumor Rep.	OC	-	-	-	-	-	-	-	-
OVKATE	JCRB	OC	Mut	-	-	-	HOMDEL	-	-	AMP
OVMANA	JCRB	OC	Mut	-	-	-	-	-	-	-
OVSAHO	Sekisui XenoTech	OC	-	-	-	AMP	-	-	-	AMP
SKOV3	ATCC	OC	Mut, AMP	-	-	-	HOMDEL	-	-	-
TOV112D	ATCC	OC	-	-	-	-	-	-	-	-
TOV21G	ATCC	OC	Mut	-	-	-	Mut	-	-	-
UWB1289	ATCC	OC	-	-	-	-	-	-	-	-

The mutations shown in this table are from cBioPortal analysis of the Cancer Cell Line Encyclopedia (CCLE) database (Broad 2019). ^1^ Weiget at al 2013 report a *PIK3R1* mutation in Ishikawa cells. EC = endometrial cancer; CC = cervical cancer; OC = ovarian cancer; Mut = mutation; AMP = amplification; HOMDEL = homozygous deletion; the dash (-) indicates absence of driver mutations.

**Table 2 cancers-16-03520-t002:** PAM inhibitors used in this study.

Drug	PAM Specificity	Cell-Free Assay Ki (nM)					
		PI3Kα	PI3Kβ	PI3Kγ	PI3Kδ	mTOR	AKT1/2/3
Gedatolisib	Pan-PI3K/mTOR	0.4	6	8	6	1	-
Dactolisib	Pan-PI3K/mTOR	4	75	5	7	6	-
Samotolisib	Pan-PI3K/mTOR	6	77	23	38	165	-
Copanlisib	Pan-PI3K	0.5	3.7	6.4	0.7	40	-
Alpelisib	PI3Kα	5	>1000	250	290	-	-
Inavolisib	PI3Kα	0.04	101.7	21.8	12.8	-	-
Capivasertib	AKT	-	-	-	-	-	3/8/8
Everolimus	mTOR	-	-	-	-	1.6	-

## Data Availability

All data are available in the main text or in the Appendix A. The datasets analyzed during the current study are available from the corresponding author upon reasonable request.

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
