# Peer review of "Functional Assessments of Gynecologic Cancer Models Highlight Differences Between Single-Node Inhibitors of the PI3K/AKT/mTOR Pathway and a Pan-PI3K/mTOR Inhibitor, Gedatolisib"

_cancers, 2024, doi:10.3390/cancers16203520_

Round 1
Reviewer 1 Report
Comments and Suggestions for Authors
The article titled "Functional Assessments of Gynecologic Cancer Models Highlight Differences Between Single-node Inhibitors of the PI3K/AKT/mTOR Pathway and a Pan-PI3K/mTOR Inhibitor, Gedatolisib" is a comprehensive examination of the efficacy of the PI3K/AKT/mTOR (PAM) pathway in gynecologic cancers. The authors check out how Gedatolisib, a pan-PI3K/mTOR inhibitor, exerts growth inhibitory and cytotoxic effects on endometrial, ovarian and cervical cancer fashions compared to different inhibitors. The authors make a strong case for focused on the PAM pathway, as it is frequently dysregulated in gynecologic cancers. The creation of gedatolisib as a pan-inhibitor that addresses the constraints of single-node inhibitors (along with drug resistance and compensatory activation of the pathway) offers a new perspective. The inclusion of extensive history facts on the complexity of the PAM pathway contributes to the clarity of the studies objective.
The look at utilizes a thorough and properly-finished methodology to assess drug efficacy. The use of multiple cancer fashions (inclusive of cellular lines and xenograft models), diverse assays (e.G., increase fee inhibition, DNA replication), and aggregate cures with palbociclib and fulvestrant emphasize the rigor of the studies. Each experiment is certainly defined and the results are statistically well established. One of the standout capabilities of this work is the distinctive evaluation among gedatolisib and single-node inhibitors which include alpelisib, capivasertib and everolimus. The use of growth rate (GR) metrics, mobile viability assays and drug synergy analyzes gives a complete review of ways multiple node inhibition plays better in gynecologic cancer models. The consequences demonstrating the advanced efficacy of gedatolisib impartial of the mutational repute of the PAM pathway are compelling. Although the item substantially discusses non-scientific data, it would be useful if there has been extra direct scientific evidence supporting the use of gedatolisib in gynecologic cancers. The article mentions initial efficacy in stable tumors, however special medical trial consequences specializing in endometrial, ovarian or cervical most cancers would toughen the relevance of this research. While the study makes development in addressing the restrictions of single node inhibition, the complex network of the PAM pathway and interactions with different signaling pathways, inclusive of the MAPK or CDK pathway, nonetheless depart questions unanswered.
Author Response
Specific comments
1) Although the item substantially discusses non-scientific data, it would be useful if there has been extra direct scientific evidence supporting the use of gedatolisib in gynecologic cancers. The article mentions initial efficacy in stable tumors, however special medical trial consequences specializing in endometrial, ovarian or cervical most cancers would toughen the relevance of this research.
[Author RESPONSE]
We have added additional details on clinical trials that tested gedatolisib in OC and EC in the Introduction (lines 126-129). A Phase 1 trial showed preliminary efficacy of gedatolisib combined with carboplatin and paclitaxel in clear cell ovarian cancer (Colombo 2021), while a Phase 2 clinical trial in recurrent EC showed that gedatolisib met clinical benefit response criteria in the stathmin-low patient subpopulation (Del Campo, 2016).
2) While the study makes development in addressing the restrictions of single node inhibition, the complex network of the PAM pathway and interactions with different signaling pathways, inclusive of the MAPK or CDK pathway, nonetheless depart questions unanswered.
[Author RESPONSE]
We agree with the reviewer that addressing the limitations of single node PAM inhibitors with a multi-node inhibitor like gedatolisib may not be sufficient to prevent potential resistance mechanism due to the activation of other compensatory pathways (e.g. CDK pathway). A comprehensive inhibition of multiple PAM pathway nodes, as well as targeting of PAM-interconnected pathways, is likely needed to minimize resistance to PAM inhibitors and increase their therapeutic efficacy. We mention this in the Introduction (lines 107-112) and in the Discussion (lines 791-797). We also used bioinformatic analyses to identify pathways potentially dysregulated (e.g. CDK, ER pathways) in association with the PAM pathway upregulation in endometrial cancer (see Figure 7). In line with these observations, our study shows that the combination with a CK4/6 inhibitor (palbociclib) or an ER inhibitor increased gedatolisib efficacy in vivo (Figures 5 and 6). While we recognize that there are still unanswered questions, we believe that the data presented in our study provide a strong, rational support to evaluate gedatolisib in combination with CDK4/6 and/or ER inhibitors in gynecologic cancers.
Reviewer 2 Report
Comments and Suggestions for Authors
In this manuscript, the authors analyzed the efficacy of pan-PAM inhibitors vs. single-node PAM inhibitors in both in vitro and in vivo models of various gynecological cancers. Further, the authors showed that pan-PAM inhibitor gedatolisib has greater efficacy than the other single node PAM inhibitors used in this study. In this study, authors have also done a series of experiments to compare gedatolisib efficacy in combination with Estrogen and CDK signaling inhibitors which is a plus. Overall, the manuscript is well written and results presented are easy to interpret for the readers. This reviewer has some minor comments on this manuscript. My Specific comments are discussed below.
Comments to the authors:
1) In Fig. 2 C, there is a range of different doses of inhibitors tested. It would be great if the doses in between lowest and highest mentioned in the figure.
2) Statistically, 2 biological independent samples were considered to make conclusions which can be accepted if the statistic test show significance. However, authors should also mention the number of replicates in each biological independent samples tested.
3) In xenograft studies, what was the rational of selecting the in vivo dose of Geda? It is shown that 2 doses tested in Ishikawa xenograft. The TGI is higher at a dose of 15mg/kg Geda in SKOV3 when compared to Ishikawa xenograft, this could be due to the aggressiveness of the cell type used. These results are consistent to the in vitro results that shown Geda has more cytotoxic effect in SKOV3 than Ishikawa cells. However, it is good for the readers if authors can provide the images of the tumor mice or excised tumor at the latest time point before harvest.
4) To further confirm the anti-proliferative or cytotoxic effect of Geda, including H&E stain of xenograft tumor sections would be helpful.
Author Response
Specific comments
1) In Fig. 2 C, there is a range of different doses of inhibitors tested. It would be great if the doses in between lowest and highest mentioned in the figure.
[Author RESPONSE]
We reported the individual drug concentrations in the legends of Figure 2C (line 434) as well as Figure 3B (lines 510) and Figure 4C (line 547).
2) Statistically, 2 biological independent samples were considered to make conclusions which can be accepted if the statistic test show significance. However, authors should also mention the number of replicates in each biological independent samples tested.
[Author RESPONSE]
Due to the large number of cell lines tested, for most experiments we only ran biological replicates as opposed to technical replicates.
3) In xenograft studies, what was the rational of selecting the in vivo dose of Geda? It is shown that 2 doses tested in Ishikawa xenograft. The TGI is higher at a dose of 15mg/kg Geda in SKOV3 when compared to Ishikawa xenograft, this could be due to the aggressiveness of the cell type used. These results are consistent to the in vitro results that shown Geda has more cytotoxic effect in SKOV3 than Ishikawa cells. However, it is good for the readers if authors can provide the images of the tumor mice or excised tumor at the latest time point before harvest.
[Author RESPONSE]
The in vivo dose of gedatolisib (15 mg/kg) was chosen based on our previous studies (Rossetti 2024, Sen 2024), tolerability in mice, and extrapolation from the gedatolisib dose used in patients (180 mg/kg). Using CDER guidelines, 180 mg administered every 7th day in humans are equivalent to 37.5 mg/kg in a mouse. However, we found that doses > 20-25 mg/kg are not well tolerated in mice. The 15 mg/kg dose that we chose for the present study is lower than the equivalent dose used in human and within the tolerability range in mice. We added a note in the methods to address this question (line 302-304).
In the Ishikawa model, we tested an additional, lower gedatolisib dose (7.5 mg/kg) to better highlight potential additive effects of the gedatolisib/fulvestrant combination. At higher concentrations of gedatolisib and/or fulvestrant, the TGI induced by the single drugs could partially mask the additive effects of the combination.
The in vivo studies were performed by Crown Bioscience. The tumors were measured periodically to calculate tumor volume, but images of the tumors were not taken. Therefore, we are unable to provide tumor images associated with the tumor growth curves.
4) To further confirm the anti-proliferative or cytotoxic effect of Geda, including H&E stain of xenograft tumor sections would be helpful.
[Author RESPONSE]
Unfortunately, tumor sections were not available to us at the time of the study, so we cannot provide H&E or other IHC stainings.